# MOCVD Grown HgCdTe Heterostructures for Medium Wave Infrared Detectors

Waldemar Gawron [1,2], Jan Sobieski [1], Tetiana Manyk [2], Małgorzata Kopytko [2], Paweł Madejczyk [2,*] and Jarosław Rutkowski [2]

1   Vigo System S.A., 129/133 Poznanska Str., 05-850 Ozarow Mazowiecki, Poland; wgawron@vigo.com.pl (W.G.); jsobieski@vigo.com.pl (J.S.)
2   Institute of Applied Physics, Military University of Technology, Kaliskiego 2, 00-908 Warsaw, Poland; tetjana.manyk@wat.edu.pl (T.M.); malgorzata.kopytko@wat.edu.pl (M.K.); jaroslaw.rutkowski@wat.edu.pl (J.R.)
*   Correspondence: pawel.madejczyk@wat.edu.pl

**Abstract:** This paper presents the current status of medium-wave infrared (MWIR) detectors at the Military University of Technology's Institute of Applied Physics and VIGO System S.A. The metal–organic chemical vapor deposition (MOCVD) technique is a very convenient tool for the deposition of HgCdTe epilayers, with a wide range of compositions, used for uncooled infrared detectors. Good compositional and thickness uniformity was achieved on epilayers grown on 2-in-diameter, low-cost (100) GaAs wafers. Most growth was performed on substrates, which were misoriented from (100) by between 2° and 4° in order to minimize growth defects. The large lattice mismatch between GaAs and HgCdTe required the usage of a CdTe buffer layer. The CdTe (111) B buffer layer growth was enforced by suitable nucleation procedure, based on (100) GaAs substrate annealing in a Te-rich atmosphere prior to the buffer deposition. Secondary-ion mass spectrometry (SIMS) showed that ethyl iodide (EI) and tris(dimethylamino)arsenic (TDMAAs) were stable donor and acceptor dopants, respectively. Fully doped (111) HgCdTe heterostructures were grown in order to investigate the devices' performance in the 3–5 μm infrared band. The uniqueness of the presented technology manifests in a lack of the necessity of time-consuming and troublesome ex situ annealing.

**Keywords:** MOCVD; HgCdTe growth; MWIR photodiodes; infrared detectors





## 1. Introduction

Many branches of science and industry require detectors operating in the medium-wave infrared (MWIR) band (3–5 μm), where the atmosphere is mostly transparent. This region is typically exploited for thermal imaging, understood as the detection and processing of slight temperature differences in many types of devices and objects, as well as in an environment. Thermal imaging is often used in astronomy and astrophysics to survey distant galaxies, whose near-light speed has shifted their emission spectra from the visible and ultraviolet to the MWIR region.

The progress in infrared (IR) detector technology has been associated mainly with photon detectors, among which photodiodes are typically the most sensitive devices. They are characterized both by a high signal-to-noise ratio and a fast response [1]. In spite of other competitive technologies and materials, mercury cadmium telluride ($Hg_{1-x}Cd_xTe$) is still the main material for infrared detectors [2].

Conventional HgCdTe IR photodetectors need to be cooled to the temperature of liquid nitrogen (77 K) in order to reduce the noise and leakage currents resulting from the thermal generation processes. Increasing the operating temperature without detectivity deterioration in so-called high-operating-temperature (HOT) detectors has been the subject of research by many scientific centers, and can be realized in a different ways [3]. Among them, we can list: a suppression of Auger thermal generation in non-equilibrium devices;

an optical immersion; multiple passes of IR radiation; magnetoconcentration detectors; Dember detectors; barrier detectors; and alternative materials such as superlattices and cascade infrared detectors [4,5]. In general, there are two main epitaxial growth techniques applied to HgCdTe: liquid-phase epitaxy (LPE); and vapor-phase epitaxy (VPE). LPE is the most technologically mature method, and has been used widely for industrial production for many years. The more advanced VPE techniques, such as molecular-beam epitaxy (MBE) and metal–organic chemical vapor deposition (MOCVD), allow the construction of more complex device structures, with good lateral homogeneity and abrupt composition and doping profiles. Historically, the best HgCdTe detectors have been grown on bulk CdZnTe substrates. However, bulk CdZnTe substrates present many technical and cost-related challenges that justify the search for a viable alternate substrate for HgCdTe growth by MOCVD—such as GaAs, for example. GaAs is an easily available, inexpensive, and high-quality substrate material, but the lattice mismatch between GaAs and the HgCdTe layer requires the use of an additional CdTe buffer layer. Modern advances in the MOCVD of HgCdTe have created the opportunity to realize novel detector designs through multilayer in situ growth, with complete flexibility in the choice of the alloy compositions and the doping concentrations [6]. This allows for the construction of complex HgCdTe heterostructure-based IR detectors dedicated to HOT conditions. In this paper, we present MOCVD-grown (111) HgCdTe heterostructures for medium-wave infrared detectors operating above 200 K. The uniqueness of the presented technology manifests in a lack of the necessity of time-consuming and troublesome ex situ annealing.

## 2. Materials and Methods

### 2.1. MOCVD-Grown HgCdTe

The HgCdTe epitaxial growth was carried out in the horizontal reactor of an AIX-200 Aixtron MOCVD (AIXTRON, Aachen, Germany) unit. The system operated in the laminar flow regime with process pressures from 50 to 1000 mbar, using a butterfly valve for pressure control. The reactor pressure of 500 mbar was used for all successful growth runs. The hydrogen was used as a carrier gas. Dimethylcadmium (DMCd) and diisopropyl telluride (DIPTe) were used as the precursors. Ethyl iodide (EI) and tris(dimethylamino)arsenic (TDMAAs) were used as donor and acceptor dopant sources, respectively. The DMCd and EI were delivered through one channel, while DIPTe and TDMAAs were delivered through another channel over the quartz container, where the elementary mercury was being held. Aixtron's gas foil rotation technique was applied for better composition uniformity. There were two temperature zones in the reactor: the mercury source zone; and the growth zone with the graphite susceptor. High-temperature annealing was used before each growth run for the reactor and the substrate cleaning. The gas delivery system was additionally equipped with Piezocon ultrasonic precursor concentration monitors. The usage of Piezocons contributed to a better repeatability of the growth processes. Adaptation of the EpiEye reflectometer (ORS Ltd., St Asaph, UK) allowed for in situ monitoring of the thickness and the surface morphology of the growing layer. Laser radiation, incident and reflected from the growing layer, with a length of 650 nm, was passed through the hole in the quartz liner, allowing us to obtain reliable interferograms. Figure 1 presents a fragment of the precursors' delivery installation, together with the horizontal reactor cell scheme with an internal mercury source of the Aixtron AIX 200 MOCVD, designed for HgCdTe deposition.

Growth was carried out on 2-inch, epi-ready, semi-insulating (100) GaAs substrates, oriented 2° off toward the nearest <110>. Typically, a 3–4-μm-thick CdTe layer was used as a buffer layer, reducing the stress caused by the crystal lattice misfit between the GaAs substrate and the HgCdTe epitaxial layer structure [7]. The buffer also prevented gallium diffusion from the substrate to the HgCdTe layer. The GaAs substrate was annealing in a Te-rich atmosphere prior to the growth of the CdTe buffer. The interdiffused multilayer process (IMP) technique was applied for HgCdTe deposition [8]. The HgCdTe was grown at 350 °C, with the mercury source kept at a temperature of 160–220 °C. The II:VI mole ratio was kept in the range of 1.5–5 during the CdTe cycles of the IMP process. Acceptor

and donor doping were examined over the wide range of the compositions, and doping levels of $5 \times 10^{14}$–$5 \times 10^{17}$ cm$^{-3}$ were obtained. In order to reduce the mercury vacancies' concentration, and to increase the uniformity of the HgCdTe heterostructures, the in situ annealing was performed under mercury-rich conditions. However, the obtained heterostructures were not annealed ex situ [9].

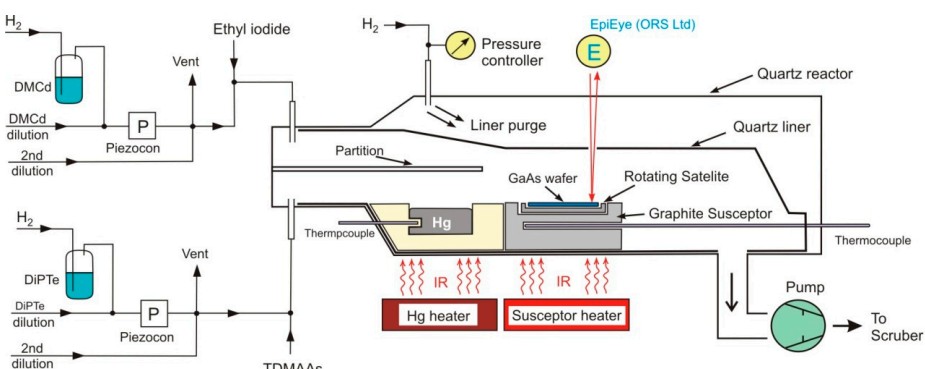

**Figure 1.** A fragment of the precursors' delivery installation, together with the horizontal reactor cell scheme with an internal mercury source of the Aixtron AIX 200 MOCVD, designed for HgCdTe deposition.

The crystalline quality of the HgCdTe films depends on the MOCVD system purity, the substrate quality, the nucleation, and the growth conditions. The growth of the HgCdTe epilayers with the (111) B orientation has some advantages, such as a lack of large macrodefects, high growth rate, lower consumption of the precursors, and effective n-type doping with iodine up to $10^{18}$ cm$^{-3}$. However, the (111) B layers exhibit some drawbacks, which affect the performance of the devices-namely, relatively rough surface morphology, high ($>10^{15}$ cm$^{-3}$) concentration of residual donor defects, and less efficient p-type doping with arsenic in comparison with (100) HgCdTe. Most of our research, and all of the work presented in this paper, concerns (111) HgCdTe; the selected growth parameters are listed in Table 1.

**Table 1.** Typical growth parameters of the (111) HgCdTe layers.

| Prior-to-Growth Annealing | 600 °C for 7 min | | | |
|---|---|---|---|---|
| Nucleation Condition | Te flush for (111) growth orientation | | | |
| **Growth Pressure** | CdTe buffer | | HgCdTe | |
| | 950 mbar | | 500 mbar | |
| Susceptor Temperature | 350 °C | | | |
| Mercury Zone Temperature | (210–220) °C | | | |
| II:VI Ratio | 1.03 during buffer growth and 1.5 during CdTe IMP cycles | | | |
| Buffer Thickness | 3–4 μm of CdTe | | | |
| Wafer Rotation Rate | 35–60 rpm | | | |
| **H$_2$ Gas Flow Rates (111) HgCdTe** | HgTe | reactor gas artery | upper | 600 sccm |
| | | | lower | 180 sccm |
| | CdTe | | upper | 1200 sccm |
| | | | lower | 1200 sccm |
| **In Situ Annealing** | Susceptor temperature | | 350 °C | |
| | Mercury zone temperature | | (180–220) °C | |
| | Time | | 15–40 min | |

### 2.2. Photodiode Design

MOCVD-grown HgCdTe heterostructures are designed for infrared photodiode construction. In this section, the fundamental rules of infrared photodiode design will be discussed. A classic $N^+/p/P^+$ structure was enriched with a transient P layer inserted between the absorber and the highly doped $P^+$ region. This additional P layer helped to shape an interface between the p and the $P^+$, preventing the formation of unfavorable potential barriers; then, a photodiode took the form of the $N^+/p/P/P^+$ structure. Furthermore, in order to improve the electrical contact properties of the metallization of the $P^+$ layer, the structure was upgraded with the $P^+/n^+$ tunneling junction. Finally, the resulting structure took the form of the $N^+/p/P/P^+/n^+$. An example of the HOT MWIR HgCdTe photodiode architecture is presented in Figure 2.

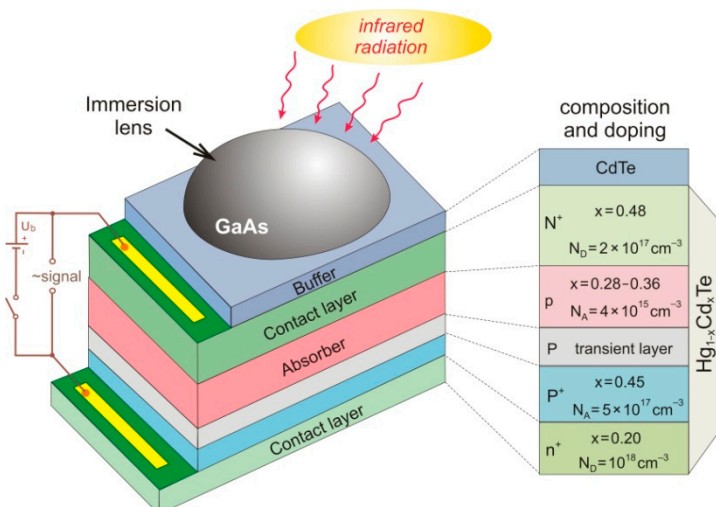

**Figure 2.** The general idea of a HOT MWIR HgCdTe photodiode with graded interfaces.

The schematic immersion lens on the top was formed from GaAs substrate. The immersion lens increases the optical area of the detector by approximately 10 times. Under the lens, there is a 3-μm CdTe buffer layer. Below the lens, there is a heavily donor-doped contact $N^+$ layer, with the composition $x = 0.48$. However, this composition of the $N^+$ layer can be changed, in the range from $x = 0.38$ to $x = 0.57$, depending on the cut-on wavelength ($\lambda_{\text{CUT-ON}}$) of the detector. For technological reasons, the thickness of the $N^+$ layer was extended to 10 μm. Next, there is a medium-doped p-type absorber region with a doping level higher than the donor-like background concentration, which is typically $3 \times 10^{15}$ cm$^{-3}$ in size. The absorber composition corresponds to the required cut-off wavelength of the detector, and for MWIR devices has a value in the range of 0.27–0.36. The thickness of the absorber should be greater than the minority carrier diffusion length, and is typically about 3 μm. The parameters of the next transient P layer were calculated individually for each device, in order to prevent the formation of a potential barrier, taking into account the interdiffusion processes during growth. The composition of the $P^+$ layer should be significantly larger than that of the absorber, and the doping should be as high as possible in order to attain low concentrations of the minority carriers and low resistance. Our MOCVD system enabled us to achieve an acceptor concentration of around $10^{17}$ cm$^{-3}$. The thickness of the $P^+$ layer was about 1 μm, in order to minimize the in-diffusion of the electrons from the metallic contact. Finally, the $n^+$ layer, with a composition lower than that of the absorber and donor concentration, at about $10^{18}$ cm$^{-3}$, finishes the structure. The thickness of the $n^+$ layer is typically about 1 μm, so as to achieve low resistance.

Following the rules described above, the obtained HgCdTe heterostructures were subjected to the standard processing procedure, which includes, among others: photolithography masking; chemical etching; metallization; cutting; and assembling. The circular photodiodes with mesa diameters ranging from 100 to 900 μm were fabricated

and mounted on the thermoelectric coolers. The structures were prepared to be back-side illuminated (irradiated from the substrate side, as was illustrated in Figure 2). The photodiodes were not passivated, and no anti–reflection coating was applied.

### 2.3. Surface Characterization

Figure 3a presents an atomic force microscopy (AFM) image of (100) GaAs oriented 2° off toward the nearest <110> epi-ready substrate, as typically used in our experiments. Although the substrate is optically perfect, its surface is not completely smooth if we consider the micro scale. The AFM micrograph shows the typical small-grain texture visible on the surface of the best commercially available, epi-ready GaAs wafer. The surface state of the substrate directly conditions the crystalline quality of the on-growing material. Figure 3b presents an AFM image of MOCVD-grown CdTe (111) B buffer on (100) GaAs 2° → <110> substrate. The (111) B growth was enforced via a suitable nucleation procedure, based on GaAs substrate annealing in a Te-rich atmosphere prior to the growth of the CdTe buffer. The buffer is 3 μm thick, and its surface roughness is typically about 10 nm; its growth on misoriented (100) GaAs substrates contributes to the reduction of the twin domain sizes [10,11]. Visible mounds and unevenness result from the existence of small amounts of the (100) phase and fast three-dimensional (111) growth [12].

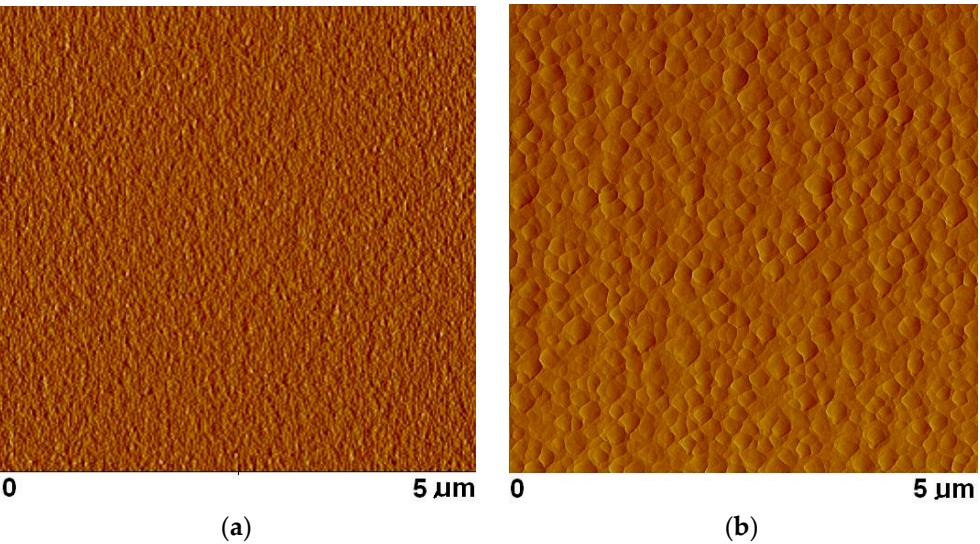

| 0 | 5 μm | 0 | 5 μm |
| (**a**) | | (**b**) | |

**Figure 3.** Examples of AFM micrographs: (**a**) GaAs(100) 2° → <110> substrate; (**b**) MOCVD-grown CdTe (111) B buffer layer.

Figure 4 presents the surface morphology of a MOCVD-grown (111) HgCdTe heterostructure, taken with a Nomarski microscope. The heterostructure is a 20-μm-thick MWIR photodiode, and its surface roughness is typically about 40 nm. The rough surface of the CdTe buffer is enhanced and reflected at the on-growing HgCdTe surface. Much effort has been made in order to optimize the growth parameters and to improve the surface quality of the (111) HgCdTe, but the results are still unsatisfactory. On the other hand, the (111) HgCdTe layers were devoid of the troublesome hillocks typical of (100) HgCdTe. A high surface roughness reflects poor crystalline quality, but nevertheless, the (111) HgCdTe heterostructures presented in this paper are good materials for high-performance HOT IR detectors, which is supported by our previous research [13].

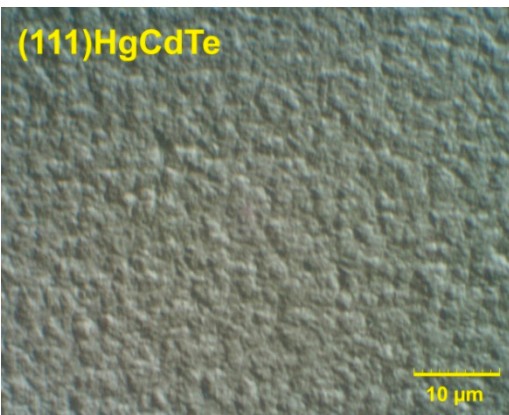

**Figure 4.** Surface morphology of a MOCVD-grown (111) HgCdTe heterosturcture obtained with a Nomarski microscope.

*2.4. Heterostructure Characterization*

Eight heterostructures of type $N^+/p/P/P^+/n^+$ were investigated during this study. They differed in the composition and the thickness of the $N^+$ and absorber regions. The details of the chosen $N^+/p/P/P^+/n^+$ heterostructures are presented in Table 2.

**Table 2.** The parameters of the chosen $N^+/p/P/P^+/n^+$ heterostructures.

| No | Composition (*x*) | | Thickness [μm] | |
|---|---|---|---|---|
| | $N^+$ Region | Absorber | $N^+$ Region | Absorber |
| #241 | 0.389 | 0.319 | 10.01 | 4.32 |
| #465 | 0.392 | 0.286 | 9.54 | 3.95 |
| #716 | 0.57 | 0.271 | 9.76 | 4.17 |
| #773 | 0.48 | 0.296 | 9.88 | 4.24 |
| #858 | 0.499 | 0.359 | 9.79 | 4.31 |
| #871 | 0.515 | 0.338 | 9.55 | 4.40 |
| #892 | 0.471 | 0.320 | 9.48 | 3.75 |
| #962 | 0.479 | 0.337 | 10.52 | 5.62 |

Figure 5 presents the scanning electron microscopy (SEM, Hitachi, Tokyo, Japan) image of the $N^+/p/P/P^+/n^+$ #962 HgCdTe heterostructure cleavage. The sample was broken along the growth direction. This allowed us to specify the individual component layers, and to measure their thickness. The obtained structure was compared with the designed one (Figure 2). The measured thicknesses were given with 0.01 μm resolution. However, because of the blurred boundaries between the layers, the practical measurement precision was assumed at about 1%, and this was sufficient for our experiments. The graded interfaces between the individual layers were also hardly observed. The measured thickness of the presented MWIR heterostructure was 20.72 μm.

Figure 6 presents the $N^+/p/P/P^+/n^+$ HgCdTe heterostructure cleavage, taken with an optical microscope at ×1000 magnification. The thickness of the CdTe buffer layer was about 2.5 μm. The total thickness of the HgCdTe structure was estimated at 20.6 μm, and was comparable with the result obtained using the SEM method. The P transition layer is barely visible, due to the interdiffusion process during growth, and because the image resolution is too low. An inset with a diagram of the individual component layers was added to the photo at the bottom for better visualization. This method is less precise than the SEM measurements, but both are prompt and useful, and give better insight into the device performance.

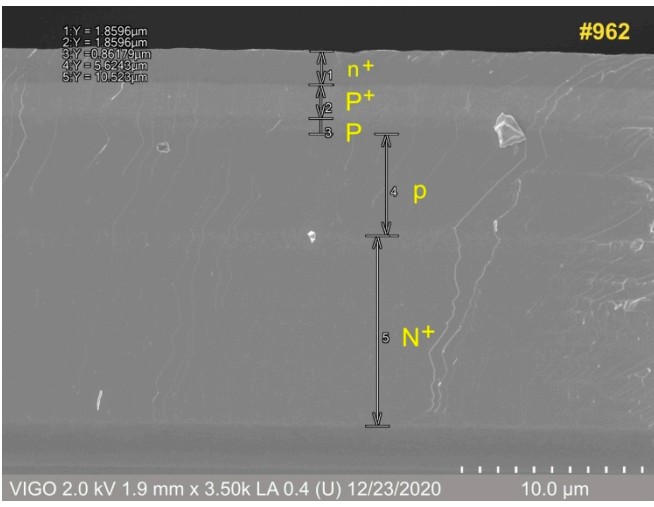

**Figure 5.** SEM image of the MOCVD-grown $N^+/p/P/P^+/n^+$ #962 HgCdTe heterostructure cleavage.

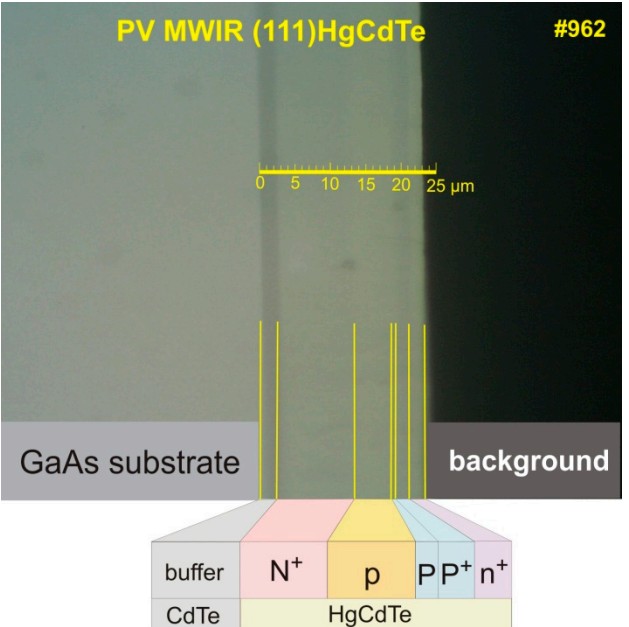

**Figure 6.** The $N^+/p/P/P^+/n^+$ HgCdTe heterostructure cleavage taken with an optical microscope.

Figure 7 presents the secondary-ion mass spectroscopy (SIMS, Cameca, Gennevilliers, France) profile of the $N^+/p/P/P^+/n^+$ $Hg_{1-x}Cd_xTe$ heterostructure with the absorbing region composition $x = 0.32$. The green line represents iodine counts: $10^5$ and $2 \times 10^4$ in the $n^+$ and $N^+$ regions, respectively, which correspond to the designed concentration levels: $10^{18}$ cm$^{-3}$ and $2 \times 10^{17}$ cm$^{-3}$, respectively. In general, abrupt transitions were observed in the iodine profile. As a positive consequence, there was no iodine presence in the adjacent layers, especially in the absorber region. The pink line represents the arsenic-stepped profile, corresponding to abruptly increasing doses of the arsenic in successive layers: p, P, and $P^+$. We did not observe any undesirable arsenic diffusion to the adjacent layers, suggesting that arsenic is a stable dopant. The mercury, cadmium, and tellurium counts enabled us to calculate a composition ($x$) profile (the light blue line) of the whole $Hg_{1-x}Cd_xTe$ heterostructure. We can see that the narrower gap absorber with the composition $x = 0.32$ is surrounded by wider energy gap $N^+$ and $P^+$ layers with compositions of around $x = 0.46$, as assumed in the heterostructure project. Due to the interdiffusive processes in the HgCdTe alloy during growth, a composition profile of the interfaces between the particular layers is not abrupt. Systematic studies of the SIMS results, supported by the theoretical calculations,

allow us to make the corrections in the programmed growth recipes, and to obtain the assumed profiles in the real HgCdTe heterostructures.

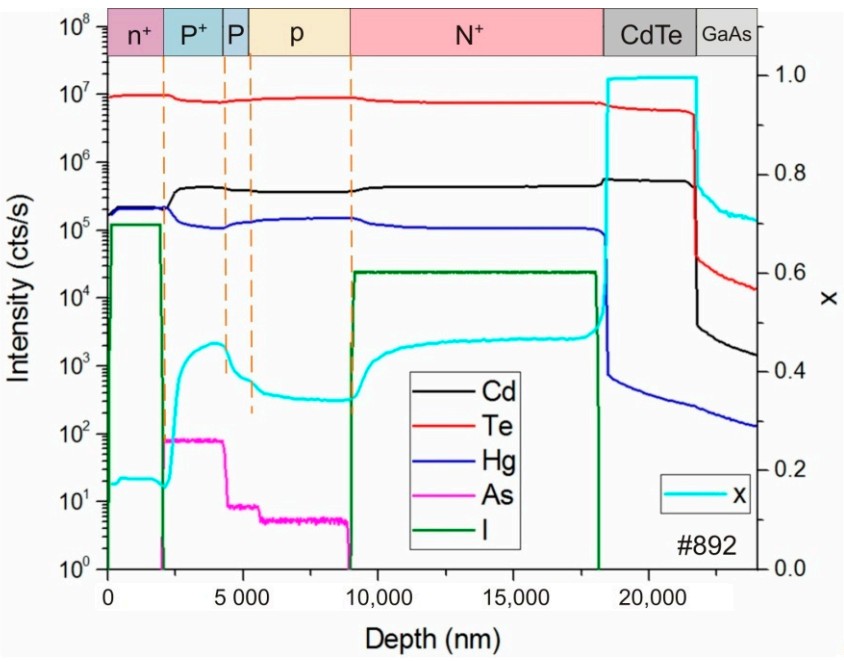

**Figure 7.** The SIMS profile of the $N^+/p/P/P^+/n^+$ Hg$_{1-x}$Cd$_x$Te #892 heterostructure with an absorbing region composition of $x = 0.32$.

Another characterization method we used was IR transmission measurement, which was performed using a PerkinElmer Fourier-transform infrared (FTIR, PerkinElmer, Waltham, MA, USA) spectrometer. Figure 8 presents the transmittance characteristics of particular fragments of the #773 $N^+/p/P/P^+/n^+$ structure, after selective etching. The selected area of the sample was covered with a photoresist film deposited during the photolithography process. As a result, the part of the sample surface intended for etching was exposed. The selective etching was performed in a solution of Br in HBr diluted in deionized water (50:50:1 Br:HBr:H$_2$). The green line ($t_a$) represents the transmission of the total heterostructure just after growth, where the narrow-gap $n^+$ top layer mainly determines its shape. Additionally, the clear interference fringes were useful in the estimation of the actual thickness of the measured sample. The red line ($t_b$) was obtained after 4-μm-deep chemical etching. As a result, the $n^+$ and $P^+$ layers were removed. The slope of the $t_b$ curve depends on the absorber transmission. This allows us to determine the composition of the absorber, which in this exemplary structure takes a value of $x = 0.302$. After removing another 7 μm of the material (11 μm in total), the $N^+$ layer was exposed, and the blue curve $t_c$ represents its transmission spectral dependence. In this way, the composition ($x$) of the obtained $N^+$ layer was determined at the value of $x = 0.492$ (whereas the predicted value was $x = 0.48$). The $N^+$ layer composition determines the $\lambda_{CUT-ON}$ value of the photodiode when it is back-side illuminated. In general, the IR transmission measurement results provide valuable feedback for MOCVD growers.

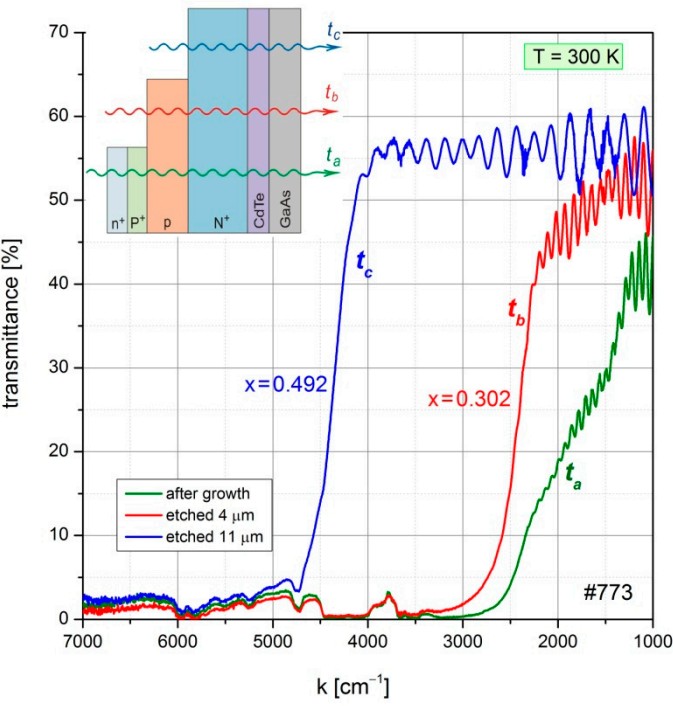

**Figure 8.** Transmittance characteristics of particular fragments of the #773 $N^+/p/P/P^+/n^+$ structure after selective etching.

## 2.5. Device Characterization

The obtained MOCVD-grown (111) HgCdTe photodiodes were characterized using the available measurement methods. The spectral characteristics were measured using a PerkinElmer Spectrum 2000 FT–IR Spectrometer. The absolute values of current sensitivity $R_i$ [A/W] were obtained using a supporting measurement set, which includes a lock-in analyzer, a monochromator, and a chopped 1000 K blackbody-calibrated silicon carbide rod. The current–voltage characteristics of the photodiodes were measured using a Keithley 2400 SourceMeter (Keithley, Cleverland, OH, USA), controlled via the LabView application for automation.

## 3. Results and Discussion

In this section, we describe the exemplary results of MWIR photovoltaic detectors fabricated with MOCVD-grown $N^+/p/P/P^+/n^+$ heterostructures. Figure 9 presents the $J(V)$ characteristics of MWIR HgCdTe photodiodes fabricated with #858 structure measured at the temperatures $T = 230$ K and $T = 300$ K. The absorber composition of this structure was designed at the value $x = 0.359$ in order to enable the MWIR operation. The circular mesa diameter of the photodiode was 820 μm. The obtained $J(V)$ characteristics follow a classic Shockley equation with the parameter $\beta = 1.1$ and shunt resistance $R_b$ ($R_b A = 5 \times 10^7$ Ω·cm²):

$$J = J_s\left(e^{qU/\beta kT} - 1\right) - \frac{U}{R_b A} \tag{1}$$

where $q$ is the elementary charge, $U$ is the voltage, $k$ is Boltzman's constant, $A$ is detector's area.

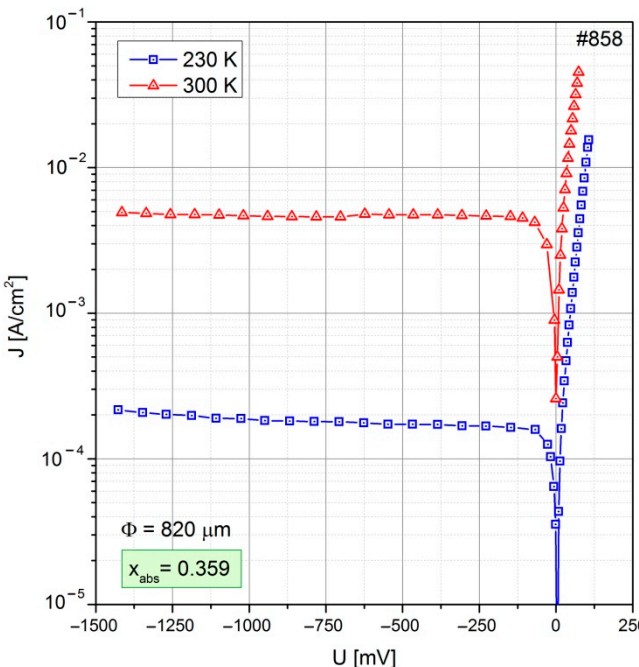

**Figure 9.** Example of the current–voltage characteristics of an MWIR HgCdTe photodiode, measured at different temperatures.

A value of the β coefficient, close to the unity, proves the predominant share of diffusion current in the total junction current. The determined value of the $R_0A$ product was 160 $\Omega\text{cm}^2$, consistent with the value determined on the basis of the saturation current determined from the fitting ($J_s = 0.136\ \text{mA/cm}^2$) for temperature $T = 230$ K. A high detector series resistance value will be a matter for future processing improvements.

Figure 10 presents the spectral characteristics of MWIR HgCdTe photodiodes of different architectures measured at the temperature $T = 230$ K. The detectors were measured without biasing voltage. The presented #773, #241, and #858 structures differ mainly in their N$^+$ the absorber compositions. The composition ($x$) of the N$^+$ layer determines the cut-on wavelength ($\lambda_{\text{CUT-ON}}$) of the detector. For example, structure #858 contains an N$^+$ layer with composition $x = 0.499$, and the $\lambda_{\text{CUT-ON}}$ is equal to 2.1 μm; this is the shortest cut-on wavelength seen in Figure 10. On the other hand, structure #241 contains the N$^+$ layer with the lowest composition ($x = 0.389$) and the $\lambda_{\text{CUT-ON}}$ is equal to 2.7 μm; this is the longest cut-on wavelength of the presented characteristics. Analogously, the cut-off wavelength ($\lambda_{\text{CUT-OFF}}$) of the detectors can vary as a result of the absorbers' composition changes. Hence, we can adjust the shape of the spectral characteristics to the needs of the recipient by appropriate shaping of the energy gap profile of the heterostructure. The maximal current sensitivity values, $R_i(\lambda_{\text{peak}})$, of the presented photodiodes are comparable, and are above 1.7 A/W at 230 K.

Table 3 presents the parameters of exemplary MWIR photodiodes, obtained on the basis of MOCVD-grown HgCdTe heterostructures. The detectivity ($D$ *) of all presented MWIR photodiodes at 230 K was above $10^{10}$ cm·Hz$^{1/2}$/W (and $10^{11}$ cm·Hz$^{1/2}$/W with an optical immersion), and was determined by background-limited photodetector (BLIP) condition, which confirms the device quality of MOCVD-grown (111) HgCdTe material.

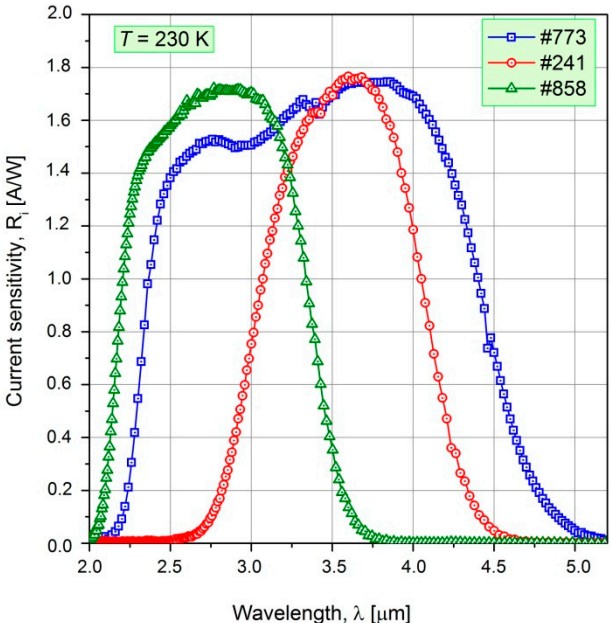

**Figure 10.** Spectral characteristics of MWIR HgCdTe photodiodes of different architectures measured at $T$ = 230 K.

**Table 3.** The parameters of the chosen MWIR photodiodes.

| No | \multicolumn{6}{c}{Photodiode Parameters at Temperature $T$ = 230 K} | | | | | |
|---|---|---|---|---|---|---|
| | $\lambda_{10\%CUT-ON}$ | $\lambda_{10\%CUT-OFF}$ | $R_0A$ | $R_i (\lambda_{peak})$ | $D * (\lambda_{peak})$ Non Immersed | $D * (\lambda_{peak})$ Immersed |
| | µm | µm | $\Omega cm^2$ | A/W | $cm \cdot Hz^{1/2}/W$ | $cm \cdot Hz^{1/2}/W$ |
| #716 | 1.8 | 5.7 | 0.7 | 1.9 | $1.23 \times 10^{10}$ | $1.23 \times 10^{11}$ |
| #465 | 2.85 | 5.2 | 1.1 | 2.0 | $1.62 \times 10^{10}$ | $1.62 \times 10^{11}$ |
| #773 | 2.2 | 4.75 | 2.4 | 1.75 | $2.09 \times 10^{10}$ | $2.09 \times 10^{11}$ |
| #241 | 2.7 | 4.35 | 1.8 | 1.78 | $1.84 \times 10^{10}$ | $1.84 \times 10^{11}$ |
| #962 | 2.18 | 3.98 | 9 | 2.1 | $4.86 \times 10^{10}$ | $4.86 \times 10^{11}$ |
| #871 | 2.0 | 3.9 | 6.5 | 2.2 | $4.32 \times 10^{10}$ | $4.32 \times 10^{11}$ |
| #858 | 2.1 | 3.6 | 160 | 1.7 | $1.66 \times 10^{11}$ | $1.66 \times 10^{12}$ |

Figure 11 shows a comparison of the $R_0A$ product of MWIR HgCdTe detectors, as a function of the 50% cut-off wavelength ($\lambda_{50\%CUT-OFF}$), with theoretical calculations. Using numerical simulations of I–V characteristics on the SimuApsys platform (Crosslight Inc., Vancouver, BC, Canada)) for a given $N^+/p/P/P^+/n^+$ heterostructure, the values of the $R_0A$ product as a function of cut-off wavelength were calculated at temperatures of 170, 200, 230, and 280 K. The numerical model implemented in the APSYS platform incorporates HgCdTe generation–recombination mechanisms: radiative, Auger, and Shockley–Read–Hall (SRH). The $R_0A$ product determined experimentally was consistent with the theoretical values for most of the samples. The dominant contribution to the dark current was from SRH recombination. The reason for the location of some experimental points slightly below or above the theoretical curve was a different value of the SRH lifetime compared to the one adopted in the calculations. Some samples were characterized by much lower values of the $R_0A$ product in relation to the theoretical calculations. This is probably due to improper processing and an increase in the share of leakage currents in the total current flowing through the junction.

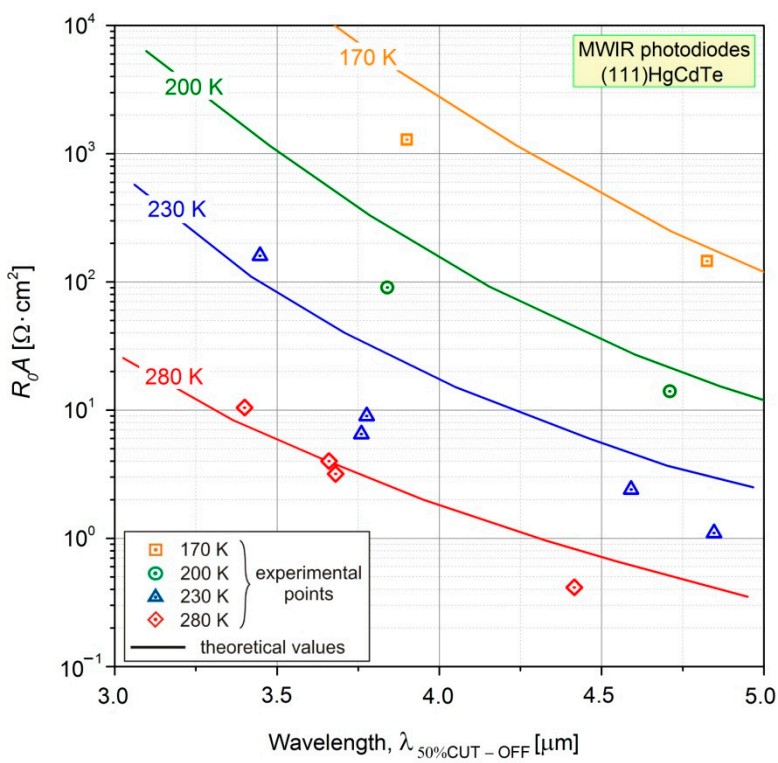

**Figure 11.** $R_0A$ data at different temperatures for MWIR HgCdTe photodiodes. Single points represent experimental values. Curves represent calculated $R_0A$ values for analyzed $N^+/p/P/P^+/n^+$ heterostructures.

## 4. Conclusions

The MOCVD technique, with a wide range of composition and donor/acceptor doping, is a very convenient tool for the deposition of HgCdTe epilayers used in uncooled infrared detectors. The unique advantage of the MOCVD growth method for HgCdTe heterostructures is the lack of the necessity of time-consuming and unsuitable ex situ post-growth annealing in sealed quartz ampoules in the presence of mercury vapors. The SIMS profile of the $N^+/p/P/P^+/n^+$ $Hg_{1-x}Cd_xTe$ heterostructure showed that EI and TD-MAAs are stable donor and acceptor dopants, respectively. Fully doped (111) HgCdTe heterostructures were grown in order to investigate the devices' performance in the 3–5 µm infrared band. The detectivity $D^*$ of all presented MWIR photodiodes at 230 K was above $10^{10}$ cm·Hz$^{1/2}$/W (and $10^{11}$ cm·Hz$^{1/2}$/W with an optical immersion), and was determined by BLIP condition, confirming the device quality of MOCVD-grown (111) HgCdTe material.

**Author Contributions:** Conceptualization, W.G. and P.M.; methodology, W.G.; investigation, J.S. and T.M.; writing—original draft preparation, P.M.; writing—review and editing, J.R.; supervision, M.K.; funding acquisition, J.R. All authors have read and agreed to the published version of the manuscript.

**Funding:** The writing of the paper has been partially achieved with the financial support of The National Centre for Research and Development (Poland)—Grant No. Mazowsze/0090/19-00; and the National Science Center (Poland)—Grant No. UMO-2017/27/B/ST7/01507.

**Institutional Review Board Statement:** Not applicable.

**Informed Consent Statement:** Not applicable.

**Data Availability Statement:** The data presented in this study are available on request from the corresponding author.

**Acknowledgments:** We would like to thank Cezary Kobyłecki from VIGO System S.A. for SEM measurements.

**Conflicts of Interest:** The authors declare no conflict of interest.

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
