# Peer review of "MOCVD Grown HgCdTe Heterostructures for Medium Wave Infrared Detectors"

_coatings, doi:10.3390/coatings11050611_

Round 1

Reviewer 1 Report

This manuscript from Gawron et al. shows their investigation of the MOCVD grown HgCdTe heterostructures on GaAs substrate for uncooled infrared detectors. HgCdTe system is a promising material for applications in uncooled infrared detectors. In this paper, the authors created HgCdTe heterostructures using the MOCVD method, and they also showed their study on the surface, material composition, device design, and IR transmittance testing.

Even though the concepts and the MOCVD grown HgCdTe heterostructures are not new, this manuscript is a complete study that covers the HgCdTe growth method, experimental setup, and detailed study of material’s fundamental properties, which are beneficial to early researchers in this field. They also show their method can bypass the necessity of time-consuming ex-situ post-growth annealing. 

In summary, I suggest publishing this paper after minor revisions in both figures and texts to improve readability, as shown in the letter to the authors below.

The manuscript by Gawron et al. investigates the MOCVD grown HgCdTe heterostructures on GaAs substrate for uncooled infrared detectors. The HgCdTe system is useful for use in uncooled infrared detectors, because of many advantages, such as tunable bandgap for the entire IR spectral range, and the ability to prepare heterostructures with complex dopings. This study intends to provide a complete picture of the growth method and testing for HgCdTe heterostructures. 

I would recommend this work for publication, but I also suggest the authors consider addressing some minor concerns, listed below.

1. Line 41 - Authors mentioned the conventional HgCdTe IR detectors need to be cooled well below ambient temperature to reduce the noise and leakage currents. While in line 55, the authors mentioned the proposed MOCVD-grown HgCdTe structures are able to operate above 200K. I suggest authors indicate the typical working temperature of the conventional HgCdTe detectors, so readers can have an idea of improving the working temperature.

2. Figure 1 - The two Piezocons in the figure are not at the same location in the input channels. One is before vent and EI input, while the other one is after the vent and TDMAAs. Can authors describe more about this configuration? 

3. The EpiEye reflectometer is mentioned in the manuscript and the diagram. The general readers may need information on how this tool works and what aspect of the information is captured by this tool. Can authors describe more about this? 

4. Based on figure 1, there is an opening at the cross of the quartz liner and the incident beam of EpiEye. Can the author comment if there is a true opening or window for this beam or not? If not, I suggest that the authors close this opening, use dash lines for the incident and reflected beam of the EpiEye, and close the gap.

5. Authors explained in detail how to grow HgCdTe with GaAs substrate, with a buffer layer (CdTe)  in between. Can authors also comment on the method of directly growing HgCdTe directly on CdTe substrate? 

https://aip.scitation.org/doi/abs/10.1063/1.97328?journalCode=apl

6. The substrate cleaning method is critical for the epi layer use the MOCVD method. Can authors add the missing information of the cleaning method to the manuscript? 

7. Line 111, “The surface state of the substrate conditions directly the crystalline…” seems missing one word after directly. Can authors double check this sentence? 

8. Line 128, the authors mentioned the obtained HgCdTe has a high roughness surface, does this surface roughness also related to the thickness? See reference paper: 

https://journals.pan.pl/dlibra/show-content?id=105834&/surface-smoothness-improvement-of-hgcdte-layers-grown-by-mocvd-klos-k-rutkowski-j-madejczyk-p-gawron-w-piotrowski-a-rogalski-a-mroz-w%3flanguage=pl

9. Figure 7 - Authors show the SIMS profile of the HgCdTe material for different elements and the composition profile. However, since this is a multi-layered structure, I suggest authors add dash lines at the 3 interfaces to show the lower composition absorber is surrounded by the higher composition of N+ and P+ layers. 

10. In table 2, the authors clearly showed the parameters of the MWIR photodiodes. I suggest authors also briefly show what differences in the sample IDs in the beginning since figures 6, 7, 8, and 9 have different sample IDs before table 2.

11. Authors mentioned the selective etching was used to prepare the sample for transmittance measurement, can authors add more information about this selective etching? 

Author Response

On behalf of all authors I would like to thank the reviewers for comments that contribute to the improvement of the final version of our paper. The manuscript has been corrected according to the reviewer comments and answers for the questions have been included.

Reviewer 1:

  1. Line 41 - Authors mentioned the conventional HgCdTe IR detectors need to be cooled well below ambient temperature to reduce the noise and leakage currents. While in line 55, the authors mentioned the proposed MOCVD-grown HgCdTe structures are able to operate above 200K. I suggest authors indicate the typical working temperature of the conventional HgCdTe detectors, so readers can have an idea of improving the working temperature.

Relevant comment. The sentence: “The conventional HgCdTe IR photodetectors need to be cooled well below ambient temperature  to reduce the noise and leakage currents resulting from the thermal generation processes.”  was amended to: “The conventional HgCdTe IR photodetectors need to be cooled to the liquid nitrogen temperature (77K) to reduce the noise and leakage currents resulting from the thermal generation processes.” - In this way we define more precisely what temperatures we are talking about.

  1. Figure 1 - The two Piezocons in the figure are not at the same location in the input channels. One is before vent and EI input, while the other one is after the vent and TDMAAs. Can authors describe more about this configuration? 

 Absolutely right !!! Obvious author’s mistake. Figure 1 was corrected.

  1. The EpiEye reflectometer is mentioned in the manuscript and the diagram. The general readers may need information on how this tool works and what aspect of the information is captured by this tool. Can authors describe more about this? 

 Adaptation of the EpiEye reflectometer allows for in situ monitoring of the thickness and the surface morphology of the growing layer. Incident and reflected from the growing layer laser radiation with a length of 650 nm passing through the hole in the quartz liner allows to obtain reliable interferograms. – These information has been added to the article.

  1. Based on figure 1, there is an opening at the cross of the quartz liner and the incident beam of EpiEye. Can the author comment if there is a true opening or window for this beam or not? If not, I suggest that the authors close this opening, use dash lines for the incident and reflected beam of the EpiEye, and close the gap.

There is a ~4mm hole in the liner over the substrate area. This hole is sufficient for in-situ reflectometry and at the same time is small enough not to cause a noticeable disturbance of a gas flow or a layer quality.

  1. Authors explained in detail how to grow HgCdTe with GaAs substrate, with a buffer layer (CdTe)  in between. Can authors also comment on the method of directly growing HgCdTe directly on CdTe substrate? 

https://aip.scitation.org/doi/abs/10.1063/1.97328?journalCode=apl

Historically, the best HgCdTe detectors have been grown on bulk CdZnTe substrates. However, bulk CdZnTe presents many technical and cost challenges that justify the search for a viable alternate substrate for HgCdTe growth by MOCVD, such as GaAs, for example. GaAs is an easily available, inexpensive and high-quality substrate material, but the lattice mismatch between GaAs and the HgCdTe layer requires the use of an additional CdTe buffer layer.

- The article has been supplemented with this information.

  1. The substrate cleaning method is critical for the epi layer use the MOCVD method. Can authors add the missing information of the cleaning method to the manuscript? 

We use epi-ready substrates from our long-term, tried and tested global substrate supplier.

Over the years we have tried various methods of cleaning the substrates, but so far we have not been able to improve the quality of the purchased substrates. We practice only prior to growth annealing as it was described in the paper.

  1. Line 111, “The surface state of the substrate conditions directly the crystalline…” seems missing one word after directly. Can authors double check this sentence? 

 The sentence:” The surface state of the substrate conditions directly the crystalline quality of the on-growing material” was amended as follow: „The surface state of the substrate directly conditions the crystalline quality of the on-growing material.”

  1. Line 128, the authors mentioned the obtained HgCdTe has a high roughness surface, does this surface roughness also related to the thickness? See reference paper: 

https://journals.pan.pl/dlibra/show-content?id=105834&/surface-smoothness-improvement-of-hgcdte-layers-grown-by-mocvd-klos-k-rutkowski-j-madejczyk-p-gawron-w-piotrowski-a-rogalski-a-mroz-w%3flanguage=pl

Right. A high surface roughness of presented layers may be related with theirs high thickness which is about 20 microns. Because structure thickness of presented structures is almost constant we do not discuss its influence on surface roughness in this research, however.

  1. Figure 7 - Authors show the SIMS profile of the HgCdTe material for different elements and the composition profile. However, since this is a multi-layered structure, I suggest authors add dash lines at the 3 interfaces to show the lower composition absorber is surrounded by the higher composition of N+ and P+ layers. 

 Dash lines were added in Figure 7 at the interfaces.

  1. In table 2, the authors clearly showed the parameters of the MWIR photodiodes. I suggest authors also briefly show what differences in the sample IDs in the beginning since figures 6, 7, 8, and 9 have different sample IDs before table 2.

 Additional table (new Table No 2) has been inserted in section “Heterostructure Characterization” with details of presented samples.

Automatically these details were cut from changed Table 3.

  1. Authors mentioned the selective etching was used to prepare the sample for transmittance measurement, can authors add more information about this selective etching? 

Selected area of the sample was covered with a photoresist film deposited in the photolithography process. As a result, the part of the sample surface intended for etching was exposed. The selective etching was performed in the solution of Br in HBr diluted in deionised water (50:50:1 Br:HBr:H2). – This information has been added to the article.

Reviewer 2 Report

That manuscript is promising, but unfortunately does not follow style of MDPI Coatings. There is no distinction between experimental sections, results and discussion. Some abbreviations are not explained. Introduction is too concise. I suggest to rewrite/revise this manuscript based on MDPI guidelines.

Author Response

Rewiever

1.That manuscript is promising, but unfortunately does not follow style of MDPI Coatings.

The manuscript was corrected following MDPI Coatings style.

2.There is no distinction between experimental sections, results and discussion.

The paper was reorganized into parts: experimental and results and discussion.

3 .Some abbreviations are not explained.

According to reviewer’s remark all abbreviations were explained.

  1. Introduction is too concise.

Introduction has been extended accordingly.

  1. I suggest to rewrite/revise this manuscript based on MDPI guidelines.

The manuscript was corrected based on MDPI guidelines.

Once again we would like to thank the reviewers for their comments.

Additional corrections:

New funding grant has been added. During our revision one Author noticed that some measurements were performed within researches covered with grant no. UMO-2017/27/B/ST7/01507.

Funding: The writing of the paper has been partially done under financial support of The National Centre for Research and Development (Poland) – the grant no. Mazowsze/0090/19-00 and the National Science Center (Poland) – the grant no. UMO-2017/27/B/ST7/01507.

Reviewer 3 Report

The author responds appropriately to the question and corrects the text, so accept it.

Author Response

Thank you,

Kind regards

Round 2

Reviewer 2 Report

The authors have improved the manuscript based reviewers comments but the characterization methods are not described in section materials & methods. For example, it is not clear which characterization method is introduced. 

Also, the some part of section material & methods belong to results & discussion. It is sometimes hard to distinguish between method and results.

Author Response

  1. The authors have improved the manuscript based reviewers comments but the characterization methods are not described in section materials & methods. For example, it is not clear which characterization method is introduced. 

The Characterization Methods section has been added to the section materials & methods:

„The obtained MOCVD grown (111)HgCdTe structures were characterized by the available measurement methods. The surface characterization of obtained layers were performed using the atomic force microscope (AFM) and Nomarski microscope. N+/p/P/P+/n+ heterostructures were characterized using scanning electron microscope (SEM) providing detailed insight into the cleavage profile. Secondary ion mass spectrometry (SIMS) revealed the doping and composition profiles of the HgCdTe heterostructures.  Fourier transform infrared spectroscopy (FTIR) provided of transmittance characteristics of the heterostructures as well as spectral responses of obtained photodiodes. The current-voltage characteristics of photodiodes were measured using a Keithley 2400 sourcemeter controlled via the LabView application for an automation.”

  1. Also, the some part of section material & methods belong to results & discussion. It is sometimes hard to distinguish between method and results.

Authors have moved appropriate part of measurement methods to the proper section in order to distinguish between the methods and results parts following the Reviewer remark.

Thank You